# Whole Sequencing and Detailed Analysis of SARS-CoV-2 Genomes in Southeast Spain: Identification of Recurrent Mutations in the 20E (EU1) Variant with Some Clinical Implications

**DOI:** 10.3390/diseases11020054

**Published:** 2023-03-31

**Authors:** María José López-Andreo, María Rosario Vicente-Romero, Enrique Bernal, Inmaculada Navarro-González, Francisco Salazar-Martínez, Vanesa Cánovas-Cánovas, Cristina Gil-Ortuño, María Gema Riquelme-Rocamora, Francisco Solano, Francisco Javier Ibáñez-López, Cristina Tomás, Carmen Candel-Pérez, Santiago Pérez-Parra, César Flores-Flores

**Affiliations:** 1Servicio de Biología Molecular, Área Científica y Técnica de Investigación (ACTI), Universidad de Murcia, 30100 Murcia, Spain; 2Laboratorio de Microbiología del Hospital General Universitario Reina Sofía de Murcia, 30003 Murcia, Spain; 3Departamento de Bioquímica y Biología Molecular B e Inmunología, Universidad de Murcia, 30100 Murcia, Spain; 4Sección de Apoyo Estadístico, Servicio de Investigación Biosanitaria, Área Científica y Técnica de Investigación (ACTI), Universidad de Murcia, 30100 Murcia, Spain; 5Hospital General Universitario Santa Lucía, 30202 Cartagena, Spain

**Keywords:** SARS-CoV-2 variants, COVID-19, concern mutations, genomic sequences, clinical implications

## Abstract

During the COVID-19 pandemic caused by SARS-CoV-2, new waves have been associated with new variants and have the potential to escape vaccinations. Therefore, it is useful to conduct retrospective genomic surveillance research. Herein, we present a detailed analysis of 88 SARS-CoV-2 genomes belonging to samples taken from COVID-19 patients from October 2020 to April 2021 at the “Reina Sofía” Hospital (Murcia, Spain) focused to variant appeared later. The results at the mentioned stage show the turning point since the 20E (EU1) variant was still prevalent (71.6%), but Alpha was bursting to 14.8%. Concern mutations have been found in 5 genomes classified as 20E (EU1), which were not characteristic of this still little evolved variant. Most of those mutations are found in the spike protein, namely Δ69–70, E484K, Q675H and P681H. However, a relevant deletion in ORF1a at positions 3675–3677 was also identified. These mutations have been reported in many later SARS-CoV-2 lineages, including Omicron. Taken together, our data suggest that preferential emergence mutations could already be present in the early converging evolution. Aside from this, the molecular information has been contrasted with clinical data. Statistical analyses suggest that the correlation between age and severity criteria is significantly higher in the viral samples with more accumulated changes.

## 1. Introduction

The COVID-19 pandemic caused by SARS-CoV-2 is an enormous global problem with very serious consequences for human health. Shortly after its appearance in December 2019, the causative pathogen was identified as a novel betacoronavirus termed SARS-CoV-2, whose genome sequence was rapidly established [1,2]. Since then, genomic analysis for identifying new mutations has shown an uneven response across the world. Around 15 million SARS-CoV-2 genome sequences have been deposited and analyzed in GISAID (www.gisaid.org (accessed on 22 February 2023)). In addition to that global database, local surveillance programs to control the COVID-19 pandemic have included genomic resources to identify new mutations of the virus. According to “the gene makes the disease” [3], new waves of the pandemic are usually associated with mutations that define the genetic profile of new variants with altered infectivity, transmissibility and capacity for immune escape [3,4].

Hence, any additional analysis of SARS-CoV-2 sequences of COVID-19 specimens sequenced in local regions and in narrow periods of time would provide useful information about subtle differences among samples classified within the same variant. There are many examples of non-synonymous mutations in the spike protein, which have been fixed in concerned variants but also found in individual samples from different virus strains and locations [3,4,5,6,7].

The importance of mutations in the spike glycoprotein (S) when establishing virus variants of concern or interest (VOCs o VOIs) is completely justified since it interacts with the ACE2 cellular receptor, initiating the viral cycle [2,8]. Two domains in the spike have special relevance: the highly exposed N-terminal domain (NTD) that mostly defines antibody epitopes [9] and the receptor-binding domain (RBD) itself; moreover, the cleavage sites for TMPRSS2 and furin [9,10] define protein processing required for viral fusion to the cell membrane and infection. In addition to these two domains and cleavage sites, the complete sequencing of genomes provides valuable information about other viral mutations that cannot be disregarded.

Regarding local surveillance programs, another important aspect should also be the confrontation between the genomic data and the clinical features of COVID-19. The information provided by full-length genome sequences of SARS-CoV-2 from COVID-19 patients not only helps to detect any new genetic variants but also might have important epidemiological and clinical implications, as it allows aspects about the evolution of the virus and the effect of identified mutations in the disease severity to be elucidated [11,12].

The objective of this study also includes an attempt to correlate the molecular information obtained in the genomic analysis with the clinical data of patients with COVID-19 in Murcia, Spain. The analysis was performed with 88 sequences of SARS-CoV-2 samples from 88 patients.

## 2. Materials and Methods

### 2.1. Patients and Sample Collection

Eighty-eight patients with a positive SARS-CoV-2 RT-PCR test and clinically established COVID-19 diagnosis were selected from October 2020 to April 2021 at the “Reina Sofía” Hospital (Murcia, Spain). Specimens from nasopharyngeal swabs were collected and submitted to RT-PCR testing for SARS-CoV-2 with standardized methods. Positive swabs in PCR were frozen at −80 °C. Anonymous clinical and laboratory data were collected by trained personnel through standardized participant (or proxy) interviews and medical record reviews. The use of nasopharyngeal swabs for virus genome sequencing was approved by the local Ethics Committee (“Comité Ético de Investigación Clínica del Hospital General Universitario Reina Sofía de Murcia”, approval Code: SALICOVID-IgG. Version V2). Statistical analysis about possible correlation with severity of disease, age and sex was completely anonymous.

We classified COVID-19 severity using a version of the World Health Organization COVID-19 Clinical Progression Scale, a commonly used ordinal scale for assessing COVID-19 severity that ranges from uninfected (level 0) to infected, from asymptomatic (level 1) to death (level 7). The severity level in the cohort of patients ranged from 2 to 7, including not hospitalized (level 2), hospitalized without supplemental oxygen (level 3), with standard supplemental oxygen (level 4), with high-flow nasal cannula or noninvasive ventilation or both (level 5), with invasive mechanical ventilation or extracorporeal membrane oxygenation-ECMO (level 6), and in-hospital death (level 7).

This study conformed to principles of the Declaration of Helsinki and the Good Clinical Practice Guidelines, and it was approved by the local Ethical Committee.

### 2.2. RNA Extraction and RT-PCR

Specimens from nasopharyngeal swabs of selected COVID-19 patients were sent to Servicio de Biología Molecular-ACTI, Univ. Murcia, Spain, for RNA extraction and whole genome sequencing. Whenever commercial kits were used, the manufacturer’s instructions were followed. RNA was extracted with the MagMAX Viral/Pathogen II (MVP II) Nucleic Acid Isolation Kit (Applied Biosystems) in a Thermo Scientific KingFisher Duo Prime instrument. During the RNA extraction procedure, MS2 phage control was not added to avoid interference with subsequent sequencing. RNA was eluted in 50 μL of elution buffer and used as the template for RT-PCR.

RT-PCR was performed to establish the Ct value of each sample, ensuring adequate RNA input in the following steps for generating cDNA from SARS-CoV-2. They were carried out with the TaqPath COVID-19 CE-IVD RT-PCR Kit (Applied Biosystems), which is a multiplexed assay that contains three primer/probe sets specific to different SARS-CoV-2-genomic regions (ORF1ab, N and S genes). Amplification was performed using an Applied Biosystems QuantStudio 5 Real-time PCR machine (96-well plates), and the Ct value considered was the mean of the Cts obtained at the three targets. For samples identified as Alpha genomes after sequencing, only two targets (ORF1ab and N gene) were detected since the Δ69–70 deletion in this variant causes negative results in the S-assay within TaqPath tests [13].

### 2.3. DNA Library Construction and Sequencing for SARS-CoV-2

We faithfully followed the protocol described by Pillay et al. [14] for generating cDNA from SARS-CoV-2 viral nucleic acid extracts and subsequently produced amplicons tiling the viral genome sequencing. It used V3 nCov-2019 primers from the ARTIC network [15]. We introduced a difference from the original procedure: in the “Tiling PCR” section, the number of cycles was 30, but adjusted to 35 when the Ct was ≥30.

The generated amplicons were used for library construction according to the reference guide of Illumina Nextera DNA Flex Library Prep (currently named Illumina DNA Prep). Sequencing was carried out on an Illumina MiSeq platform with a read length of 300 bp (paired–end 150), using a MiSeq Reagent Kit v3 (600 cycles). Under these conditions, the average percentage of bases with a Q-Score ≥ Q30 was 97% (value calculated by Sequencing Analysis Viewer v2.4.7 software of Illumina). The Q-score is a prediction of the probability of a wrong base call. For Q30, this probability is 1/1000.

### 2.4. Sequencing Data Analysis

Raw reads from Illumina sequencing were assembled using Illumina DRAGEN COVID Lineage App 3.5.3. that uses a customized version of the DRAGEN DNA pipeline (detailed information on which can be found at: https://support-docs.illumina.com/SW/DRAGEN_v40/Content/SW/DRAGEN/GPipelineIntro_fDG.htm (accessed on 22 February 2023)). This software utilizes a kmer-based alignment algorithm to detect the presence of a subset of tiled 32-mers within the reference genome across all reads in the sample. It allows alignment to a reference genome, calls variants, and generates a consensus genome sequence. Next, it performs lineage/clade analysis using Pangolin (version 3.1.7) and NextClade. A detailed description of all the mutations found with regard to the reference genome was finally obtained. We always kept the reference genome that the application used by default and provided built-in support: SARS-CoV-2 (NCBI Reference Sequence: NC_045512.2) plus human controls. We obtained a percentage of sequenced bases close to 100% (average value of 99.2% in the 88 sequenced genomes) and very high sequencing coverage with a median value well above 1000×. All of the sequences were deposited in GISAID (https://www.gisaid.org/ (accessed on 22 February 2023)) (Accession IDs in Appendix A).

### 2.5. Statistical Analysis of the Data

The free software statistical package R was used to process the information and create the data matrix [16]. A descriptive and correlational study was carried out using Spearman’s correlation coefficient (ρ) for quantitative and qualitative variables. The results were interpreted according to the following scale: ρ < |0.1|: negligible effect; |0.1| < ρ < |0.3|: small effect, |0.3| < ρ < |0.5|: medium effect; ρ > |0.5|: high effect.

To explore relationships among nominal variables, Pearson’s chi-squared test was used, as the requirements about independence of data and values greater than 5 were met. The strength of the relationships was estimated using Cramer’s V index. For scalar variables, the one-way ANOVA test was chosen after checking the assumptions of normality with the Shapiro–Wilk test and homoscedasticity with Fligner–Killen’s test. For post-hoc analysis, multiple comparisons were performed with Bonferroni correction. Throughout the analysis, two-tailed tests, *p*-values < 0.05 and significance level α = 0.05 were used.

## 3. Results

### 3.1. Description of the SARS-CoV-2 Variants Detected

The full-length genome sequencing and analysis of the 88 SARS-CoV-2 samples from COVID-19 patients revealed a predominant variant identified as clade 20E (EU1) (63 genomes 71.6%; Table 1). This result is consistent with the location (Murcia, Spain) and the period (October 2020–April 2021) in which the study was conducted since the 20E (EU1) variant initially expanded in Spain in early summer 2020 and spread widely across western Europe, becoming the most prevalent [17]. Sampling was distributed along the period as detailed in Appendix A.

The 20E (EU1) variant bears a D614G change in the spike protein, which was associated with higher transmissibility and viral infectivity soon after the beginning of the ongoing pandemic [3]. Although the disease severity of its ancestral strain was not significantly affected by the mentioned change, it was proposed that its presence offers an advantage due to the increase in replication and transmission [18,19,20]. In fact, this change was a hallmark of all current, most recently evolved variants and delimitates the founding of the B.1 lineage [3]. However, during the first weeks of the epidemic, this substitution was less frequent in Spain than in other countries [21]. In addition to D614G, at that time, 20E (EU1), also known as B.1.177, was a barely modified variant with just another change in the spike protein, A222V, that did not seem to confer any advantage to the virus on the ability to mediate viral entry (Figure 1) [17].

Our results indicate that the Alpha variant (lineage B.1.1.7) was the second most prevalent (13 genomes, 14.8%; Table 1). This variant was first identified in September 2020 in the UK [22] and was one of 5 VOCs established by the WHO. It was prevalent until July 2021, particularly in Europe and North America (https://covariants.org/ (accessed on 22 February 2023)). The 13 Alpha variant genomes corresponded to samples taken between February and April 2021, and none corresponded to samples collected during the first 4 months of the trial (October 2020–January 2021). However, in Spain, this concern variant was first identified around October 2020 and became prevalent in February 2021 (https://covariants.org/per-country (accessed on 22 February 2023); http://covidtag.paseq.org/ (accessed on 22 February 2023)). The reason that the Alpha variant spread quickly was its transmissibility, which was 43–90% higher than the Wuhan strain [3,22,23]. In comparison to the whole country, our local data indicated that the spread of the Alpha variant in Murcia showed some delay.

Alpha bears several mutations in the NTD and RBD of the spike (Figure 1), which potentially affects viral function and could explain its successful spread at that time. Among these amino acid changes, it is worth noting Δ69–70, Δ144–145 and N501Y [9]. In addition, mutation P681H, close to the furin cleavage site, could facilitate the processing of the spike protein and the entry of the virus into the cell [10,24]. Interestingly, emergence mutation E484K, characteristic of many variants (some of them VOCs), was also identified in some samples classified as variant B.1.1.7 in February 2021 [5]. This E484K substitution compromises the efficacy of antibodies in neutralizing this variant [7].

Six of the sequenced genomes in our study (6.8%) belonged to lineage B.1.221 (Table 1), as they showed the single mutation that characterizes this variant: S98F (variant 20A/S:98F) on the B.1 lineage background (Figure 1). It mostly occurred in Belgium, but its track was gradually lost from February 2021 due to its low prevalence (https://covariants.org/variants/S.S98F (accessed on 22 February 2023)).

In addition, in our study, we identified 1 VOC (Gamma, lineage P.1 or B.1.1.28.1) and 1 VOI (Mu, lineage B.1.621), represented by one genome each (1.14% each, Table 1). The Gamma variant was announced in December 2020, particularly associated with Manaus, Brazil. Our Gamma sample was collected on April 12, 2021, although the presence of this variant in different parts of Europe was corroborated months later. Gamma shows multiple mutations, notably the triad K417T, E484K and N501Y located in the RBD. This particular combination can likely cause a more significant decrease in neutralization by antibodies than any mutation by itself and increases affinity of the S protein for ACE2, improving infectivity [3]. Interestingly, that triad also appears in Beta (B.1.351) and currently Omicron (BA.1, BA.2, BA.4&5, BA.2.12.1, BA.2.75, BQ.1 and XBB) variants but with some differences at position 417 (K417N) in both variants and at position 484 (E484A) in Omicron (https://covariants.org/shared-mutations (accessed on 22 February 2023)).

The Mu variant (B.1.621) was also identified in one sample collected on 12 April 2021. That variant seemed to come from South America, predominantly Colombia, in early 2021. Similar to many other B.1 lineages, including all those found in our study (B.1.177; B.1.1.7; B.1.221 and B.1.1.28.1), this variant also bears the D614G change. Moreover, it carries a constellation of mutations with functional significance, highlighting E484K and N501Y in RBD and P681H close to the cleavage site by furin (https://covariants.org/variants/21H.Mu (accessed on 22 February 2023)). 

To complete the 88 genomes, 4 (4.6%) formed a miscellaneous group without particular interest (Table 1). Two of these genomes belonged to the B.1 lineage with just the D614G change, and the other two belonged to sublineages B.1.36.10 and B.1.499, each bearing an additional mutation, T572I or S673T, respectively.

### 3.2. Considerations of Concern on the Variant 20E (EU1) Background

We carried out a detailed comparison of the 88 sequenced genomes with the double aim: (a) to identify subtle differences among those classified within the same lineage and further projections and (b) to explore possible correlations between mutations of concern or interest and the severity of disease. From the first point of view, we found diverse mutations on the spike protein in 16 of the 63 genomes classified as 20E (EU1) (Table 2), in addition to the canonical D614G and A222V reported elsewhere. In 4 of those 16 genomes, we found relevant changes that should be considered of concern as judged by the numerous evidence accumulated throughout the COVID-19 pandemic [3,4,9,25]: they are Δ69–70, E484K, Q675H and P681H. E484K is possibly the most paradigmatic example of a recurrent emergence mutation since it has been identified in very distant geographical locations and different periods without any apparent relationship [4].

To start with, we detected the E484K substitution in 2 samples (RS40 and RS73) belonging to variant 20E (EU1) (Table 2). As mentioned above, E484K is a critical mutation in the RBD of spikes. At least 3 of the 5 VOCs display changes at position 484. It appears in the Gamma and Beta variants. In Omicron, the substitution occurs in the same position, but glutamic changes to alanine instead of lysine (E484A). In February 2021, Public Health England announced that a number of Alpha cases carried this E484K substitution [5]. Regarding VOIs, an increase in the number of SARS-CoV-2 genomes with this substitution was reported in the US as soon as December 2020. Special attention was deserved for those belonging to the Iota variant (lineage B.1.526) [9,26]. Other VOIs also independently acquired E484K, such as Mu (B.1.621), Eta (B.1.525), and the variant Zeta or lineage P.2. Finally, the B.1.617.1 (Kappa) and B.1.617.3 sublineages and ancestral B.1.617 showed the E484Q substitution, functionally similar to E484K [27]. 

The presence of this change in addition to the two 20E (EU1) defining variant mutations (A222V and D614G) strongly suggest that the position 484 change was one of the first stages towards the establishment of variants with improved virus functions that appeared later on across the world. In addition, our samples shared two additional substitutions: A701G and P1069S. It is worth mentioning that position 701 was also changed in the Beta and Iota variants, but with a different amino acid (A701V). 

The double deletion Δ69–70 accompanying E484K in our RS40 sample (Table 2) is also interesting. Δ69–70 is located within the antibody-binding footprints on the NTD of the spike protein. That deletion was defined as a feature of the Alpha variant, where several reports have suggested a role in efficient spike processing, viral transmissibility and infectivity [6,9,28], but others have linked the deletion to a slightly lower viral susceptibility to convalescent plasma in immunocompromised patients [29]. The Omicron variant (BA.1, BA.4&5 and BQ.1) also bears this deletion, as well as the aforementioned substitution at position 484 (E484A) (https://covariants.org/shared-mutations (accessed on 22 February 2023)). Taken together, it is clear that this recurrent deletion evolved spontaneously in several SARS-CoV-2 lineages.

The RS34 genome is another example of variant 20E (EU1) with an additional relevant mutation in the spike: P681H substitution. This is characteristic of Alpha, Omicron and Mu variants. The isosteric P681R change appeared in Delta and Kappa (https://covariants.org/shared-mutations (accessed on 22 February 2023); https://covariants.org/variants/21B.Kappa (accessed on 22 February 2023)). Position 681 is located near the cleavage point by furin (sequence 681–685), so any change around it has raised interest. In turn and in addition to being associated with the protease recognition site, the change has also been associated with increased ACE2 affinity and the immune-evasive phenotype of the full spike [30,31]. The RS34 genome also carries an additional S:T716I mutation, which is shared by the Alpha variant. There is no evidence that this change may confer any advantage to the virus.

RS23 is another genome of the group of 20E (EU1) variants showing a change near the furin-cleavage site, the substitution Q675H. This is reminiscent the Q677H and Q677P changes described in some strains of variant Eta [3] and in multiple independent lineages circulating in the US [32,33]. After those reports, a study identified H655Y as the most significant substitution associated with increased viral fitness [4]. In the Omicron variant, its particular cluster of mutations near the furin cleavage site (H655Y, N679K and P681H) has been also associated with its increased transmissibility [30]. The accumulation of changes suggests a convergent or parallel evolution of the virus at a “hot-spot” region that surely confers an advantage in spread or viral transmission. 

Twelve remaining genomes identified as 20E (EU1) showed additional mutations without well-established relevance. We highlight the T95I change (found in genomes RS48, RS56 and RS63), which has emerged independently in 30 lineages and is strongly associated with increased viral fitness [4]. Moreover, the T20N substitution (genome RS50) that appears in the Gamma variant might confer resistance against some antibodies [34]. Finally, other mutations that lie in the antigenic supersite loop of the NTD (S254P in the RS38 and RS52 genomes) could be candidates to contribute to the immunescape of the virus [35], but this has not been contrasted in the current variants. 

Concerning other variants identified in our research, the 13 Alpha variants found maintained the characteristic pattern of mutations in the spike protein with minor variations. The only noteworthy addition would be the presence of the substitution S:A871V in 4 of the 13 Alpha genomes identified, but we have not found any reference about the possible relevance of this amino acid change in the literature. 

Aside from the spike protein, a defining mutation of the 20E (EU1) variant is A220V in the nucleocapsid protein (https://covariants.org/variants/20A.EU1 (accessed on 22 February 2023)) of SARS-CoV-2. In our study, this substitution was present in 55 of the 63 genomes classified as 20E (EU1). Interestingly, samples RS40 and RS73 presented a different homologous bulky hydrophobic substitution at the same position (A220F).

Examination of mutations in other non-surface viral proteins should not be overlooked, as they have important and diverse functions. On this side, we have identified an interesting three amino acid deletion at ORF1a in three 20E (EU1) genomes (RS40, RS73 and RS67, Table 2): S3675, ORF1a:G3676 and ORF1a:F3677. This deletion occurs at 106–108 positions of non-structural protein 6 (NSP6). NSP6 is related to restricting autophagosome expansion and double-membrane vesicle formation [3], and it has been proposed that changes in this protein compromise the ability of autophagosomes to deliver viral components to lysosomes for degradation [36]. Obermeyer et al. [4] recently reported that, within ORF1, the highest concentration of fitness-associated mutations was found in NSP4, NSP6 and NSP12–14. According to them, this deletion seems to be usual, as has been reported in 4 of the 5 VOCs, Alpha, Beta, Gamma and Omicron, as well as some VOIs (Eta, Iota and Lamba) and other circulating variants (https://covariants.org/variants (accessed on 22 February 2023)). In our old samples, RS40 and RS73 show great interest due to their previously described pattern of additional mutations in the spike. The third one (RS67) has an additional substitution at this protein (S:F192L) but with unknown relevance.

The total number of single nucleotide variants, deletions and insertions on the whole genome and the spike gene for the 88 SARS-CoV-2 sequenced genomes is shown in Appendix A. It is interesting to observe how the most recently evolved variants in our study (alpha, gamma and mu) present a higher percentage of mutations in the spike gene than in the complete genome, something that does not occur in 20E (EU1) and 20A/S:98F.

### 3.3. Correlations between Clinical Data and Genome of SARS-CoV-2 Variants

Another of the objectives of this study was to collate the information obtained in the genomic analysis (identified variants and mutations) with the clinical data of the COVID-19 patients, mostly gender, age and disease severity, in an attempt to establish correlations despite the relatively moderate number of samples. Disease severity is expressed on the WHO scale that ranges from 1 to 7 (data at Appendix A), with 1 being “not hospitalized with resumption of normal activities” and 7 being “death”.

First, we focused on the 63 samples of the 20E (EU1) variant to check whether the presence of additional mutations in the spike protein, in addition to the referred defining mutations of the variant described above, could be associated with the disease severity. In this group, the cohort’s median age was 69 (23–92) years, and the female gender comprised 43%. We used Spearman’s coefficient to assess the correlation or the statistical dependence between the rankings of our variables (presence or absence of additional mutations with gender, age and disease severity) taken two by two. For the 63 genomes, we detected a moderate but significant association between age and disease severity (Spearman’s ρ of 0.42, Figure 2a). 

The correlation between the increase of disease severity and age was still moderate but significant when considering only the 47 cases without additional mutations (Spearman’s ρ = 0.34; Figure 2b). However, it showed a high and significance value when considering the 16 cases with additional mutations (Spearman’s ρ = 0.82; Figure 2c). Within this group of 16 genomes, we established a small subset of 5 genomes bearing mutations of concern in the spike protein or a relevant deletion in ORF1a (RS23, RS34, RS40, RS67 and RS73, Table 2). Despite the higher relevance of these mutations, the severity of the disease in older patients did not seem to increase (Spearman’s ρ = 0.56; still high correlation but not significant; Figure 2d).

When the 13 genomes of the Alpha variant were incorporated into the study of statistical correlations and the cohort was 76 cases (63 20E (EU1) + 13 Alpha genomes [median age: 64 (17–92); female proportion, 45%], the association between disease severity and age remained with a high and significant value of Spearman’s ρ of 0.53. This value raised to 0.73 when considering just the 13 cases of the Alpha variant (Figure 3a,b).

We also established three groups according to their mutational backgrounds to look for possible significant differences among the regarding gender or disease severity criteria. Pearson’s chi-squared criteria were used. These three groups comprised: (1) 20E (EU1) without additional mutations in the spike (47 cases), (2) 20E (EU1) with additional mutations in the spike (16 cases) and (3) Alpha variant (13 cases). However, no correlation between the nominal variables gender and mutational background was found (χ^2^(2) = 0.53, *p* > 0.05 (0.7659) (Appendix A). We also grouped the patients into two categories to facilitate analysis and subsequent interpretation. The first category included those patients with a disease severity factor between 2 and 3, considered not critical, and the second group with a severity value greater than 3, considered clinically severe disease. When applying Pearson’s chi-square, we did not find a relationship between the two nominal variables (χ^2^(2) = 5.44, *p* > 0.05 (0.06575) (Appendix A).

Finally, we used the one-way ANOVA test for scalar variables (age versus mutational background). We found differences between groups, with F (2, 73) = 4.29, *p* < 0.05, ɳ2 = 0.105. In a post-hoc analysis, we performed multiple comparisons with the Bonferroni correction and obtained significant differences between the 20E (EU1) cases without additional mutations in the spike and the Alpha variant group despite the low number of genomes available (Appendix A).

## 4. Discussion

The aims of this study were to analyze the molecular data obtained from the sequencing of 88 SARS-CoV-2 genomes isolated in patients from October 2020–April 2021 from several points of view. First, a detailed study of variants and mutations of concern or relevance was performed in relation to the spread of the virus at that time in Europe, the possible routes they reached to Murcia, and finally the relationship to changes found in more recent variants. Second, we explored the possible correlation of the molecular data with the clinical information of the infected COVID-19 patients. It is assumed that the identification of SARS-CoV-2 variants in a specific population and period provides interesting homogeneous epidemiological information, and the detailed study of mutations can shed light on the later evolution of the virus.

From an epidemiological point of view, this study covers samples of the third European wave, which in the Murcia region was the most harmful, with above 51,000 positives and almost 900 deaths over a population of one and a half million people. At that time, the population was not yet vaccinated and since the first and second waves had low local incidence, its seroprevalence and immunological memory against SARS-CoV-2 were likely very low. 

The identified variants show some local peculiarities, with projections to the future variants appearing later during the rise of new variants. The 20E (EU1) variant or B.1.177 lineage was prevalent in Spain until approximately February 2021, when Alpha displaced it. According to https://covariants.org/per-country (accessed on 22 February 2023) (enabled by data from GISAID) from February 08 to 22, 2021, 1174 sequences of Alpha (frequency 0.49) and 888 of 20E (EU1) (frequency 0.37) were counted. Our data are consistent with that distribution, since the 13 genomes identified as Alpha correspond to samples taken in February–April 2021 in parallel with 14 genomes of the 20E (EU1) variant that were also identified in that period. Alpha was first identified in Spain around October 2020, and between December 21 and January 04, 2021, 397 sequences of this variant had already been counted (frequency 0.24) compared to 993 of 20E (EU1) (frequency 0.61) (https://covariants.org/per-country (accessed on 22 February 2023)). The absence of Alpha genome from October 2020 to January 2021 in the Murcia region seems to indicate that this variant spread throughout that region with some delay, so it took longer for B.1.177 lineage displacement.

On the other hand, the frequency of 20A/S:98F or B.1.221 lineage, which became a very abundant variant in Belgium, was always very low in Spain during the period of our study, never exceeding the value of 0.01 (https://covariants.org/per-country (accessed on 22 February 2023)). In Murcia, we found a significant percentage of 6.8% (6/88) for this variant, concentrated in January 2021 (5 of 6). The epidemiological link might be related to the presence of colonies of Belgian citizens around Torrevieja, a tourist town on the Mediterranean coast close to Murcia, even in winter. A total of 105 direct flights operated from the international airport of the Region of Murcia to Belgium during this 6 month period. 

Later on, the detection of the Gamma and Mu variants confirmed the ability of VOCs and VOIs to spread rapidly and reach areas far away from their place of origin (Sud Africa, South America).

It is assumed that the independent acquisition of identical or closely related mutations with recurrent emergence suggests converging evolution of the virus and provides cues to establish their impact on viral fitness. One of the most remarkable conclusions of this study is the early identification of concern mutations in some genomes of a variant as little evolved as 20E (EU1). The early detection of changes, such as E484K and P681H, and the deletion (Δ69–70) in the spike protein of these genomes is interesting in relation to the ulterior importance of these substitutions in the variants that appeared after the 20E (EU1) one. These substitutions have been reported in many later VOCs, VOIs and other multiple lineages circulating independently, and the selective advantages that they imply for virus fitness have been confirmed by numerous evidence throughout the pandemic [3,9,25]. The identification of the S:Q675H substitution is also relevant, as very similar mutations near the furin-binding pocket were later associated with the increased transmissibility of the Omicron variant [30].

Concerning the likely correlation between the referred spike protein mutations and the sensitivity of SARS-CoV-2 to mRNA vaccine-elicited antibodies and/or contribution to selected SARS-CoV-2 variants to spike antigenicity indicate that both recurrent deletions [6,28], E484K [7] and P681H [30] single mutations, drive antibody escape. 

It is also worth mentioning the detection of ORF1a:S3675, ORF1a:G3676, ORF1a:F3677 deletions in three of the 20E (EU1) genomes of our group, since this deletion is shared by 4 ulterior VOCS. The viral advantages associated with this deletion are still a subject of research. Overall, the 5 mutations that we identified in some of our viral genomes strongly support that they have been relevant to facilitate the evolution from the early stages of the virus, when the first variants began to appear and spread. It remains difficult to assess precisely the extent to which additional mutations may contribute to viral advantages. On the one hand, it was not possible to assess the full complexity of the virus genome since there is very little published information on mutations that affect non-structural proteins, and data about spike proteins are very predominant.

From a clinical point of view, statistical analysis shows that the correlation between age and severity criteria is significantly higher in genome groups with a higher mutational background, which is additional mutations in the spike of 20E (EU1) or Alpha self-mutations. It is obvious that we cannot sequence and analyze a huge number of SARS-CoV-2 samples, so the statistical approach is rather a search for trends with a robust conclusion to extrapolate. The small number of sequenced genomes limits the value of the analysis, although we have detected some trends that deserve further study. It is also clear that beyond viral infection, multiple factors can contribute to disease severity, such as age and other underlying comorbidities. Our results support the importance of these factors in combination with more transmissible and infective viral variants. 

## 5. Conclusions

We performed a comparative analysis of 88 SARS-CoV-2 genomes obtained from anonymous COVID-19 patients of different gender and age at the “Reina Sofía” Hospital (Murcia, Spain) in the October 2020-April 2021 period to explore the viral evolution during the early stages of the COVID-19 pandemic in relation to the appearance of the prominent variants of concern or interest appeared later. The first conclusion was that during that period, we found that, similarly to other European countries and regions, 20E (EU1) was the major variant found in Murcia, but Alpha VOC was just bursting. The second conclusion was that characteristic mutations reported in posterior VOCs, including the still current Omicron, were already detected in some (5) genomes classified as 20E (EU1). Most of those mutations are found in crucial domains of the spike protein, namely Δ69–70, E484K, Q675H and P681H. These findings confirm that preferential emergence mutations were already present in the early viral evolution since they confer advantages in infectivity or transmissibility to the virus. This also account for those mutations become prevalent during viral evolution. Statistical analyses to correlate mutations with clinical data suggest that its correspondence with age and severity criteria is significantly higher in the viral samples with more accumulated changes, but it is clear that the number of samples is not large enough to definitively conclude that intuitive point or identify the most harmful mutation at the spike protein.

## Figures and Tables

**Figure 1 diseases-11-00054-f001:**
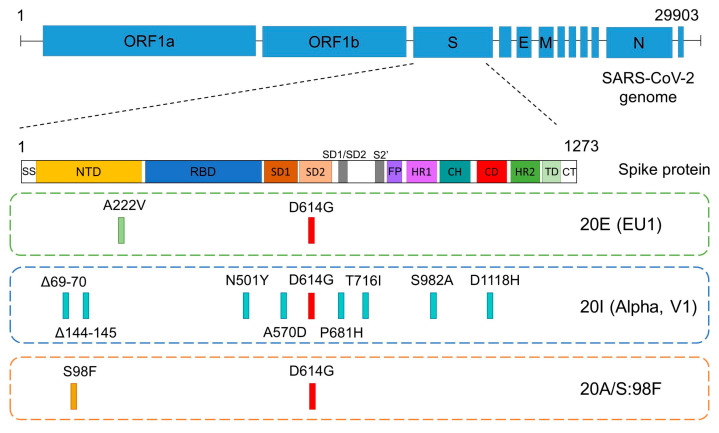
Schematic representation of the spike gene, showing its domains and the positions of the defining mutations in the 3 most prevalent SARS-CoV-2 variants of this study: 20E (EU1), Alpha and 20A/S:98F. The common D614G mutation (in red) delimitates the founding of the B.1 lineage. SS: signal sequence, NTD: N-terminal domain, RBD: receptor binding domain, SD: subdomain, FP: fusion peptide, HR: heptad repeat, CH: central helix, CD: connector domain, TD: transmembrane domain, CT: cytoplasmic tail.

**Figure 2 diseases-11-00054-f002:**
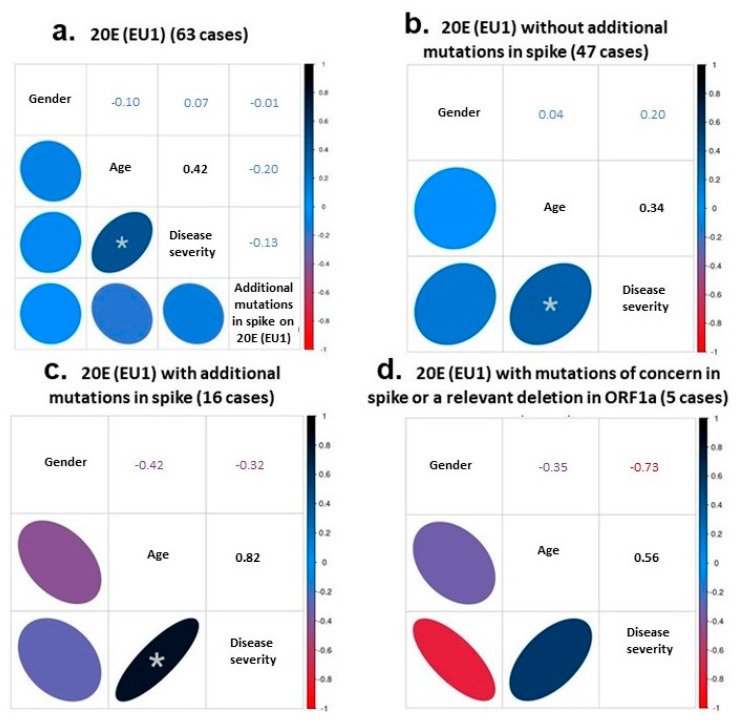
Correlational studies using Spearman’s coefficient. The statistical dependence between the rankings of gender, age, disease severity and mutational hallmark was analyzed in the 63 samples of the 20E (EU1) variant (**a**). The correlation was also investigated between gender, age and disease severity in the cases without (**b**) or with (**c**) additional spike mutations, and in the subset of genomes with relevant mutations (**d**). The Spearman’s coefficients obtained in each case are indicated in the corresponding box. An asterisk indicates a significant correlation between age and disease severity in (**a**–**c**) with *p*-values of 0.0006, 0.0211 and 0.0001, respectively.

**Figure 3 diseases-11-00054-f003:**
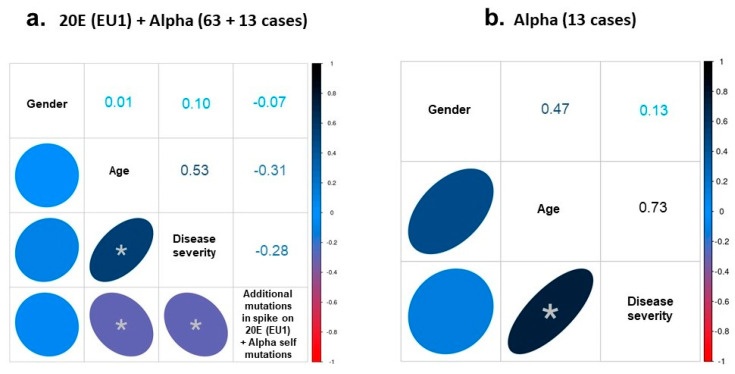
Spearman´s analysis. (**a**) Correlation between the rankings of gender, age, disease severity and mutational hallmark was analyzed in the 76 samples of the 20E (EU1) plus Alpha variants. (**b**) The correlation was also analyzed between gender, age and disease severity in the Alpha genomes. The Spearman’s coefficient obtained in each case is indicated in the corresponding box. Asterisk indicates a significant correlation with: (**a**) *p*-value of 0.0001 (age/disease severity), 0.0072 (age/additional mutations) and 0.0135 (disease severity/ additional mutations); (**b**) *p*-value of 0.0046 (age/disease severity). However, only the correlation between age and disease severity showed a high effect (Spearman’s coefficient > |0.5|).

**Table 1 diseases-11-00054-t001:** Summary of SARS-CoV-2 variants found in the study. * GISAID: the Global Initiative on Sharing Avian Influenza Data (https://gisaid.org/ (accessed on 22 February 2023)); ** WHO: World Health Organization (https://www.who.int/ (accessed on 22 February 2023)).

Nextstrain Clade	Pango Lineage	GISAID * Clade	WHO ** Name/ Status	%	Nonsynonymous Defining Mutations in Spike
20E (EU1)	B.1.177	GV	-	63/88: 71.6%	A222V, D614G
20I (Alpha, V1)	B.1.1.7	GRY	Alpha/ VOC	13/88: 14.8%	Δ69–70, Δ144–145, N501Y, A570D, D614G, P681H, T716I, S982A, D1118H
20A/S:98F	B.1.221	G	-	6/88: 6.8%	S98F, D614G
20A or 20C (miscellaneous group)	B.1, B.1.36.10 or B.1.499	GH	-	4/88: 4.56%	D614G
20J (Gamma, V3)	P.1 or B.1.1.28.1	GR	Gamma/ VOC	1/88: 1.14%	L18F, T20N, P26S, D138Y, R190S, K417T, E484K, N501Y, D614G, H655Y, T1027I, V1176F
21H	B.1.621	GH	Mu/ VOI	1/88: 1.14%	T95I, Y144S, Y145N, R346K, E484K, N501Y, D614G, P681H, D950N

**Table 2 diseases-11-00054-t002:** Genomes 20E (EU1) with additional mutations in the spike protein. D614G and A222V are the specific signature substitutions for lineage B.1 and derived sublineage B.1.177, respectively. Additional changes are indicated in italics, and among them are those of concern in bold. The presence of a 3 amino acid deletion in ORF1a is also noted, as it should be considered relevant since it appears in 4 of the 5 VOCs and some VOIs.

Sample Name	GISAID Accession ID	Mutations in Spike	Presence of ORF1a: S3675-, G3676- and F3677-Deletion
RS23	EPI_ISL_11623867	A222V, D614G, ***Q675H***	No
RS25	EPI_ISL_11634150	*T19R*, A222V, D614G	No
RS26	EPI_ISL_11634151	*S98F*, A222V, D614G	No
RS34	EPI_ISL_11635219	A222V, D614G, ***P681H***, *T716I*	No
RS38	EPI_ISL_11646217	A222V, *S254P*, *A262S*, D614G, *P809S*	No
RS40	EPI_ISL_11685202	***Δ**69–70***, A222V, ***E484K***, D614G, *A701G*, *P1069S*	**Yes**
RS48	EPI_ISL_11696722	*T95I*, A222V, D614G, *A1020S*	No
RS50	EPI_ISL_11696733	*T20N*, A222V, D614G	No
RS52	EPI_ISL_11696749	A222V, *S254P*, *A262S*, D614G, *P809S*	No
RS56	EPI_ISL_11728097	*T95I*, A222V, D614G, *A1020S*	No
RS62	EPI_ISL_11755017	A222V, *P251H*, D614G	No
RS63	EPI_ISL_11755804	*T95I*, A222V, D614G, *A1020S*	No
RS67	EPI_ISL_11760681	*F192L*, A222V, D614G	**Yes**
RS69	EPI_ISL_11765429	A222V, D614G, *N658D*	No
RS73	EPI_ISL_11767045	A222V, ***E484K***, D614G, *A701G*, *P1069S*	**Yes**
RS101	EPI_ISL_11794379	*M153I*, A222V, D614G	No

## Data Availability

All datasets presented in this study are included in the article/Appendix A.

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
