# Peer review of "Whole Sequencing and Detailed Analysis of SARS-CoV-2 Genomes in Southeast Spain: Identification of Recurrent Mutations in the 20E (EU1) Variant with Some Clinical Implications"

_diseases, 2023, doi:10.3390/diseases11020054_

Round 1

Reviewer 1 Report

The work addresses a modest SARS-Cov-2 genomic surveillance report in a short-radius geographical location (a single hospital in Spain) and on a very small sample size (N = 88), which certainly impeded authors to achieve more robust conclusions. Due to the theme importance, as well as the scope of this journal, this report might still be of interest to a Covid/Virology-related audience, after some critical adjustment from my comments below.

A-1. Lines 23-24: Please adjust "which the 20E (EU1) variant was prevalent (71.6%) but the Alpha bursting 24 in (14.8%)", as it doesn't read well.

M-1. Lines 130-137: Authors should provide detailed information on embedded tools and their respective parameters used during this DRAGEN pipeline execution. Since it's a licensed software owned by Illumina, many readers, especially data analysts/bioinformaticians who rely on open source tools, won't have access to DRAGEN and its functioning setup. This is crucial for the community to benefit from following your data analysis methods through more accessible routes rather than a licensed software. Genome Assembly and Variant Calling steps are the most critical ones for such methods description and reproducibility.

R-1. Table 1: Please add a legend on the bottom of the table for the acronyms on both "GISAID clade" and "Who Status" columns. In addition, I strongly encourage authors to prepare a schematic rectangular representation of the spike gene showing its domains and the position of the mutations from the three most prevalent variants depicted on the table. A color legend for common and specific mutations among the three variants would also be useful. That would be the new Fig.1 from the paper.

R-2. Line 172: One must never used "more evolved" terminology. Can it be replaced by 'most recently evolved'? If not, please try another word or expression.

R-3. Line 175: Same as for the comment right above. Try to replaced 'poorly evolved' to something else (e.g. 'less prevalent').

R-4. Line 330: "should not be" or "should be". Please clarify and adjust if needed.

R-5. Line 346: Please rephrase this sub-section title. I suggest removing "On the possible"

R-6. Line 368: On Figure 1 caption, the actual p-value threshold must be informed. The same

applies for Figure 2.

R-7. Supp Figs. Captions of both supplementary figures need adjustments. FigS1, I believe authors meant so say p-value > 0.05, rather than r > 0.05. Also, a bracket must be closed after "(gender versus mutational background". FigS2 needs to describe the p-value of the significant result.

R-8. Regarding Table S1, it would be interesting/important to add the total number of significant nucleotide variants found (SNVs and Indels, at least) along with its percentage on the total genome size, as well as the same kind of report for within the spike gene only. Authors may fee

l free to create another table (S2) for such report

Author Response

Reviewer 1

Thank you for providing your time and efforts for reviewing our article. Since each of your comments has important insight, we answered for them point by point (in red)

 The work addresses a modest SARS-Cov-2 genomic surveillance report in a short-radius geographical location (a single hospital in Spain) and on a very small sample size (N = 88), which certainly impeded authors to achieve more robust conclusions. Due to the theme importance, as well as the scope of this journal, this report might still be of interest to a Covid/Virology-related audience, after some critical adjustment from my comments below.

We thank to the reviewer for the general comment concerning the interest of our work. 88 is not a high sample size, but we analyzed as much cases as we could.

 A-1. Lines 23-24: Please adjust "which the 20E (EU1) variant was prevalent (71.6%) but the Alpha bursting 24 in (14.8%)", as it doesn't read well.

Right. Thank you for noting that point at the abstract. The sentence has been rewritten to facilitate the meaning according to our English advisor.

 M-1. Lines 130-137: Authors should provide detailed information on embedded tools and their respective parameters used during this DRAGEN pipeline execution. Since it's a licensed software owned by Illumina, many readers, especially data analysts/bioinformaticians who rely on open source tools, won't have access to DRAGEN and its functioning setup. This is crucial for the community to benefit from following your data analysis methods through more accessible routes rather than a licensed software. Genome Assembly and Variant Calling steps are the most critical ones for such methods description and reproducibility.

To improve methods description, we have included in the new manuscript version a link to the DRAGEN DNA pipeline, in which it is possible to find all the detailed information. In addition, we have described how the kmer-based alignment algorithm works and indicated the used pangolin version (see paragraph 2.4 at the amended manuscript).

R-1. Table 1: Please add a legend on the bottom of the table for the acronyms on both "GISAID clade" and "Who Status" columns.

Thank you for your suggestions. First, all acronyms have been defined on the bottom of the table

In addition, I strongly encourage authors to prepare a schematic rectangular representation of the spike gene showing its domains and the position of the mutations from the three most prevalent variants depicted on the table. A color legend for common and specific mutations among the three variants would also be useful. That would be the new Fig.1 from the paper.

As suggested by the reviewer, we have also added a new Figure 1 complementary to the Table, showing all defining mutations at the different spike domains from the three most prevalent variants in our study. Common D614G mutation is indicated in red. We hope this would cover your request.

R-2. Line 172 (now 181): One must never used "more evolved" terminology. Can it be replaced by 'most recently evolved'? If not, please try another word or expression.

Your suggestion is fine for us. Thank you !

R-3. Line 175 (now 184): Same as for the comment right above. Try to replaced 'poorly evolved' to something else (e.g. 'less prevalent').

We have replaced the expression by “barely modified”. We prefer that than “less prevatent”. That variant was not evolved at that date, but according to Table 1, it was prevalent indeed.

 R-4. Line 330 (now 349): "should not be" or "should be". Please clarify and adjust if needed.

We think that should not be overlooked was right. S protein is the protein that accumulated more amino acid changes, but other mutations outside the S protein cannot be ignored. In fact, we did not find a lot in our samples, but mutations in other proteins could be important for virus infectivity.

R-5. Line 346 (now 370): Please rephrase this sub-section title. I suggest removing "On the possible"

Sure !. This expression has been removed for clarity.

R-6. Line 368 (now 385): On Figure 1 (new Figure 2) caption, the actual p-value threshold must be informed. The same applies for Figure 2 (new Figure 3).

We have now incorporated the p-values to all Figures related to statistical analysis in all cases of significant correlations (see new version of the manuscript).

R-7. Supp Figs. Captions of both supplementary figures need adjustments. FigS1, I believe authors meant so say p-value > 0.05, rather than r > 0.05. Also, a bracket must be closed after "(gender versus mutational background". FigS2 needs to describe the p-value of the significant result.

Done it!

R-8. Regarding Table S1, it would be interesting/important to add the total number of significant nucleotide variants found (SNVs and Indels, at least) along with its percentage on the total genome size, as well as the same kind of report for within the spike gene only. Authors may fee l free to create another table (S2) for such report.

Done it! The total number of single nucleotide variants, deletions and insertions on the whole genome and the spike gene for the 88 SARS-CoV-2 sequenced genomes is shown in a new Table S2. To this regard, it is wort noting that the most recently evolved variants in our study (alpha, gamma and mu) present a higher percentage of mutations in the spike gene than in the complete genome in comparison to 20E (EU1) and 20A/S:98F.

Reviewer 2 Report

The manuscript of López-Andreo et al. explores the genomic makeup of 88 SARS-CoV-2 variants collected from COVID-19 patients exhibiting clinical profiles with varying levels of severity that were admitted to the Reina Sofia Hospital of Murcia in Spain between October 2020 and April 2021. This period coincided with the third COVID-19 European wave and rise of several Variants of Concern (including Alpha), which the authors detect in their sampling. Over 70% of viral variants belonged to the Nextstrain Clade 20E with the typical A222V and D614G mutations of the spike and only ~16% were VOCs or Variants of Interest (VOIs). Remarkably, a more detailed analysis of the genetic makeup of the large 20E group revealed that a significant number of variants contained a range of additional mutations, many of which manifested at later time in the current VOC Omicron variants (especially the 69-70 deletion, and the E484K and P681H of the spike protein). The authors conclude these additional mutations represent a process of 'recurrent emergence' of mutations that impact viral fitness.in addition to mutant constellation analysis the authors explore correlations between disease severity, gender an age for the 20E variants with or without additional mutations in the spike.Interestingly, larger mutations sets appear to increase severity.

The manuscript adds valuable information of variant diversity associated with a locality in Spain at the time when initial VOCs were arising during the pandemic.

Line 334. Please confirm deletion in NSP6 are in positions 105-107 and not 106-108

Figure. 1 Perhaps add p-values for correlations in individual cells in parentheses below correlation values.

Author Response

Thank you for providing your time and efforts for reviewing our article. Since each of your comments has important insight, we answered for them point by point (in red)

The paper of María José López-Andreo et.al. presents experimental and computation analysis of the molecular data obtained from the sequencing of 88 SARS-CoV- 2 genomes isolated in patients during October 2020 – April 2021 from Murcia Spain.  

The main important contribution of the presented research is the early identification of selected mutations in some genomes of a variant 20E (EU1).

The early detection of substitutions E484K and P681H and the deletion (Δ69-70) in the spike protein of these genomes are presented as being of importance for the future variants appeared after the 20E (EU1).

Another important characterized substitution emerged from this study is the S:Q675H since very similar mutations near the furin binding pocket have been later associated with the increased transmissibility of the Omicron variant.

There are some correlations presented in the manuscript, remarkably to mention being the ones that analyze the correlation across the age, gender, severity of disease and the mutations burden and the type of mutations.  However, the authors recognized in the discussion section the difficulty of accessing a larger sample size for statistical analysis, and as such, the genome sequencing data are limited, leading to less statistical power and presentation of only of some trends in the correlation statistics.  

I am recommending that the paper undergoes one more round of editing for language and a minor revision as mentioned below:

Thank you very much for the introductory comments summarizing the main contributions we have made in this article. According to your request, the English has been edited again and some grammatical errors have been corrected,

Even though the population from Murcia was not fully vaccinated and thus the serological data are not available for this study it would be of importance for the readers to add one extra correlation between the strength of antibody titers and neutralization capacity vs the mutations reported in this study. 

The serological data can be collected from other studies involving the presented mutations and deletions, not necessarily related to the population from Murcia.  It would be of interest to know whether these reported changes in the COVID19 genomes are already known to be correlated to a better or less efficient antibody titers and neutralizing capacity in other population from Spain or other countries...

This minor addition to this paper will increase the impact of the presented research on the readers from the field of molecular virology, immunology and molecular pathology of diseases.

Your suggestion is thoughtful. You are right that at that stage Murcia population was not fully vaccinated. We cannot provide with local serological data for direct discussion the sequenced samples. However, it is true that most references and studies over the world have found a correlation between the referred spike protein mutations and the decrease in the sensitivity of SARS-CoV-2 to mRNA vaccine-elicited antibodies. The contribution of SARS-CoV-2 variants to spike antigenicity indicate that both, recurrent deletions and single mutations, somehow drive antibody escape, although in different degree. Ref. 28, 6, 7 and 30 in our manuscript discussed this correlation for several recurrent deletions at the NTD domain, and for the particular  H69/V70 deletion, E484K and P681H respectively. A short sentence about this point has been added to the discussion to address your query (lines 493-496).

Reviewer 3 Report

The paper of María José López-Andreo et.al. presents experimental and computation analysis of the molecular data obtained from the sequencing of 88 SARS-CoV- 2 genomes isolated in patients during October 2020 – April 2021 from Murcia Spain.  

The main important contribution of the presented research is the early identification of selected mutations in some genomes of a variant 20E (EU1).

The early detection of substitutions E484K and P681H and the deletion (Δ69-70) in the spike protein of these genomes are presented as being of importance for the future variants appeared after the 20E (EU1).

Another important characterized substitution emerged from this study is the S:Q675H since very similar mutations near the furin binding pocket have been later associated with the increased transmissibility of the Omicron variant.

There are some correlations presented in the manuscript, remarkably to mention being the ones that analyze the correlation across the age, gender, severity of disease and the mutations burden and the type of mutations.  However, the authors recognized in the discussion section the difficulty of accessing a larger sample size for statistical analysis, and as such, the genome sequencing data are limited, leading to less statistical power and presentation of only of some trends in the correlation statistics.  

I am recommending that the paper undergoes one more round of editing for language and a minor revision as mentioned below:

Even though the population from Murcia was not fully vaccinated and thus the serological data are not available for this study it would be of importance for the readers to add one extra correlation between the strength of antibody titers and neutralization capacity vs the mutations reported in this study. 

The serological data can be collected from other studies involving the presented mutations and deletions, not necessarily related to the population from Murcia.  It would be of interest to know whether these reported changes in the COVID19 genomes are already known to be correlated to a better or less efficient antibody titers and neutralizing capacity in other population from Spain or other countries...

This minor addition to this paper will increase the impact of the presented research on the readers from the field of molecular virology, immunology and molecular pathology of diseases.

Author Response

Thank you for providing your time and efforts for reviewing our article. Since each of your comments has important insight, we answered for them point by point (in red)

The manuscript of López-Andreo et al. explores the genomic makeup of 88 SARS-CoV-2 variants collected from COVID-19 patients exhibiting clinical profiles with varying levels of severity that were admitted to the Reina Sofia Hospital of Murcia in Spain between October 2020 and April 2021. This period coincided with the third COVID-19 European wave and rise of several Variants of Concern (including Alpha), which the authors detect in their sampling. Over 70% of viral variants belonged to the Nextstrain Clade 20E with the typical A222V and D614G mutations of the spike and only ~16% were VOCs or Variants of Interest (VOIs). Remarkably, a more detailed analysis of the genetic makeup of the large 20E group revealed that a significant number of variants contained a range of additional mutations, many of which manifested at later time in the current VOC Omicron variants (especially the 69-70 deletion, and the E484K and P681H of the spike protein). The authors conclude these additional mutations represent a process of 'recurrent emergence' of mutations that impact viral fitness.in addition to mutant constellation analysis the authors explore correlations between disease severity, gender an age for the 20E variants with or without additional mutations in the spike. Interestingly, larger mutations sets appear to increase severity.

The manuscript adds valuable information of variant diversity associated with a locality in Spain at the time when initial VOCs were arising during the pandemic.

Thank you very much for the comments summarizing the main contributions we have made in this article on our work and retrospective observations concerning Sars-Cov-2 pandemic in Murcia during first waves.

Line 334 (now 353). Please confirm deletion in NSP6 are in positions 105-107 and not 106-108

Absolutely right. The deletion comprises the tripeptide SGF, positions 106-108 at NSP6. We are sorry for the shift, as 105-107 correspond to LSG at that protein. The error has been corrected (line 353). 

Figure. 1 Perhaps add p-values for correlations in individual cells in parentheses below correlation values.

Note that this is now Figure 2 and 3. We have now incorporated the p-values to all Figures related to statistical analysis in all cases of significant correlations (see new version of the manuscript).

Reviewer 4 Report

This paper is a whole genome analysis using clinical samples of SARS-CoV-2 in southeastern Spain. It is worthwhile to find important findings, such as the late epidemic in the region compared to Europe. On the other hand, despite the title of whole genome analysis, most of the actual analysis focuses on the Spike protein, so a comprehensive analysis including other regions may be required. Therefore, major revision is necessary before publication in this paper at present.

Comments;

1) In Figure 1, the numbers and letters in the figure are too small to see and should be of a larger size.

2) In Figures 1 and 2, although the age distribution of the sample in this study is unknown, some correlation between disease severity and age appears not to be unique to this region. Also, in Figure 1, the fact that there was no significant difference when narrowing down to 5 cases may be due to the small number of samples.

Author Response

Thank you for providing your time and efforts for reviewing our article. Since each of your comments has important insight, we answered for them point by point (in red). Previously to the specified replies, we think that research design and clarity of presentation have been improved after the modifications and additions of new information at the amended manuscript. This is due to the comments of the three reports. See the revised version of the manuscript for details.

 This paper is a whole genome analysis using clinical samples of SARS-CoV-2 in southeastern Spain. It is worthwhile to find important findings, such as the late epidemic in the region compared to Europe. On the other hand, despite the title of whole genome analysis, most of the actual analysis focuses on the Spike protein, so a comprehensive analysis including other regions may be required. Therefore, major revision is necessary before publication in this paper at present.

You are right. The title is referred to whole genome analysis as we sequenced the whole genome of all the samples indeed (see Table 1S at suppl. Material including the % of non-N bases in whole genome). More information concerning  the whole genome analysis has been introduced as required by Reviewer 1 (new Table 2S). However, most of the mutations were detected at the S protein encoded region. The spike protein has received special attention in our study and many other due to the relevance for infections, either for ACE2 binding as for antibodies recognition and immune response. However, please note that four paragraphs of the manuscript have been devoted to mutations in other viral proteins. Three of them are at lines 344-348, 349-363, 496-505, and a new fourth paragraph related to the new Table 2S has been added (lines 364-368). We emphasize at the manuscript (line 349) that examination of mutations in other non-surface viral proteins should not be overlooked as they can fulfill diverse functions. Finally, we inform you that other minor mutations were observed in some samples, such as the ORF3a:V256- (deletion 26158-26162 in the nucleotide sequence), but we did not report on this data in the manuscript due to the minor interest according to the current knowledge on Sars-Cov2 evolution and sequence repositories.      

Comments;

1) In Figure 1 (now Figure 2), the numbers and letters in the figure are too small to see and should be of a larger size.

Right. Done it. The size has been increased to facilitate the reading of the labels and Spearman coefficient values. 

2) In Figures 1 and 2, although the age distribution of the sample in this study is unknown some correlation between disease severity and age appears not to be unique to this region.

These are now Figures 2 and 3. The age of the patients is given at Table 1S. Anyway, about the general concept, the reviewer is absolutely right. Some statistically significant correlation between severity and age was found (see Figures 2 and 3) and that correlation is not surprising. It is true that other papers have already observed it.

Also, in Figure 1, the fact that there was no significant difference when narrowing down to 5 cases may be due to the small number of samples.

Surely, the reviewer is also right on this comment. We would like to note that In spite of the small number of samples from the statistical point of view, the obtention of whole data for each sample has been a huge effort for us, and we could not collect more reliable samples to increase the number of cases. 

Round 2

Reviewer 4 Report

None